# Genome-Wide Selection Sweep Analysis to Identify Candidate Genes with Black and Brown Color in Tibetan Sibu Yaks

**DOI:** 10.3390/ani14172458

**Published:** 2024-08-24

**Authors:** Xinming Wu, Lu Xu, Haoyuan Zhang, Yong Zhu, Qiang Zhang, Chengfu Zhang, Guangxin E

**Affiliations:** 1College of Animal Science and Technology, Southwest University, Chongqing 400715, China; wxm155156@163.com (X.W.); lujiusym@163.com (L.X.); swuzhanghy@163.com (H.Z.); 2Institute of Animal Husbandry and Veterinary Medicine, Tibet Academy of Agriculture and Animal Husbandry Science, Lhasa 850009, China; w0w777@163.com (Y.Z.); tibetzq@126.com (Q.Z.); tibetzcf@126.com (C.Z.)

**Keywords:** yak, pigment, coat color, wide-genome selection, ROH

## Abstract

**Simple Summary:**

We performed a genome-wide selection sweep analysis on 50 domesticated yaks to investigate the genetic basis of coat color phenotypes. We identified numerous candidate genes (e.g., *PLCB1*, *LEF1*, and *DTNBP1*) that play key roles in melanin synthesis. Unique candidate genes for different coat color groups may explain the variations in coat color. Additionally, a range of selected genes associated with melanin and pigment deposition were identified within the ROH islands. This study provides a solid theoretical basis for understanding yak coat color.

**Abstract:**

Although coat color is an important economic phenotype in domesticated yaks (*Bos grunniens*), its genetic basis is not yet fully understood. In this study, a genome-wide selective sweep and high-frequency runs of homozygosity (ROH) identification were performed on 50 yaks with different coat colors to investigate candidate genes (CDGs) related to coat color. The results suggested that 2263 CDGs were identified from the 5% interaction windows of the F_ST_ and θπ ratio, along with 2801 and 2834 CDGs from black and brown yaks with iHS, respectively. Furthermore, 648 and 691 CDGs from black and brown yaks, which were widely enriched in pathways related to melanogenesis, melanocyte differentiation, and melanosome organization via GO and KEGG functional enrichment, respectively, were confirmed on the basis of the intersection of three parameters. Additionally, the genome of brown yaks presented more ROH, longer ROH fragments, and higher inbreeding levels than those of black yaks. Specifically, a large number of genes related to melanin synthesis and regulation (e.g., *UST*, *TCF25*, and *AHRR*) from the ROH islands were confirmed to be under strong selection. In summary, the results of this study enhance the understanding of the genetic basis for determining yak coat color.

## 1. Introduction

Yaks (*Bos grunniens*) are a distinct livestock species native to the Qinghai-Tibet Plateau, and they play a crucial role in the agricultural economy of plateau pastoral areas [1,2]. Yaks provide various resources to local herdsmen, including dairy products, beef, leather and fur, transportation, and even fecal fuel. These animals are also deeply intertwined with local culture, folklore, mythology, entertainment sports, and Tibetan Buddhism [3,4]. With the advent of whole-genome sequencing and multiomics technology, numerous candidate genes (CDGs) associated with economic growth traits and high-altitude adaptability in yaks have been identified [3,5]. Coat color, an essential physiological phenotype for animal selection and domestication, is closely associated with adaptive evolution (e.g., protection from natural enemies and predation, courtship behavior, and resistance to UV rays) and has significant implications for livestock production [6]. Among the local breeds in Tibet, the Sibu yaks are particularly noteworthy. This breed comprises two subgroups with the following different coat colors: black and brown. Although there is no documented or scientific evidence supporting the notion that brown Sibu yaks possess exclusive advantages, local herders consider brown yaks to be superior beef producers compared with black yaks, especially considering their relatively smaller population size, making them highly sought after by herders in the region.

The determination of coat color relies on the type of pigment particles produced by melanocytes and the migration of these particles [6]. The origin of melanocytes can be traced to neural crest cells, which migrate to the epidermis and hair follicle matrix during embryonic development [7]. Melanin, which is oxidized and catalyzed by enzymes such as tyrosine kinases, can be classified into eumelanin (black or brown) without sulfur atoms and pheomelanin (red or yellow) with sulfur atoms. The relative amounts and distributions of these pigments determine the coat and skin color of animals [8]. The synthesis of melanin can be broken down into two main steps. The first involves the production of dopaquinone through the catalytic or redox production of the substrate tyrosine. In the second step, dopaquinone reacts with glutathione or cysteine to produce either pheomelanin or eumelanin, depending on the conditions [9,10]. In addition, the production of 5,6-dihydroindole-carboxylic acid (DHICA) can also generate eumelanin through the action of enzymes such as tyrosinase-related protein 1 (TYRP1) and tyrosinase-related protein 2 (TYRP2) [11]. However, it is important to note that brown coat color is not solely determined by eumelanin. The overexpression of the *SLC7A11* gene, for example, increases the ratio of phenylalanine to eumelanin, resulting in brown patches in transgenic sheep [12]. A combination of factors, such as agouti signaling protein (ASIP), phenylthiourea, and additional cysteine supplements, can also lead to the production of yellow-brown granular melanocytes [13]. The recessive homozygosity of the *TYRP1* gene is known to produce brown fur in many animals [14,15,16], whereas mutations in the melanocortin 1 receptor (*MC1R*) gene can result in color changes, such as chestnut, red, black, and brown, in animal fur [17,18,19].

Moreover, various signaling pathways have been identified as regulators of melanocyte development and, consequently, the coat color phenotype. These include the Wnt signaling, endothelin 3-endothelin receptor B (EDN3/EDNRB) signaling, and KIT/KITL signaling pathways [20,21,22]. Additionally, several CDGs involved in pigment deposition have been discovered. These genes are responsible for regulating melanin synthesis (e.g., *MC1R*, *ASIP*, and *TYR*) and melanosome morphogenesis (e.g., *MART1*, *GPNMB*, and *TYRP*) [6,23,24]. In yaks, multiple gene variations, such as *MC1R*, *PMEL*, and *KIT*, have been confirmed to be associated with the coat color phenotype [25,26,27].

With the advancement of genome sequencing technology, numerous physiological phenotypes and adaptive traits of animals have been investigated via whole-genome selection signal analysis. Methods such as the F_ST_, θπ ratio and integrated haplotype score (iHS) are often used to detect selective features of artificial domestication and adaptive evolution of populations, such as coat color, plateau adaptability, and growth traits [28,29,30]. In this study, we aimed to elucidate the genetic basis of coat color phenotypes in a single breed of yak by utilizing genome-wide selection sweep analysis to identify CDGs associated with coat color.

## 2. Materials and Methods

This study was approved by the Institutional Animal Care and Use Committee (IACUC) of Southwest University (Permit No. IACUC-20190815-04). Venous blood samples were collected from 50 adult Sibu yaks that were unrelated to each other within three generations. The samples included 25 black-coated yaks (Figure 1A) and 25 brown-coated yaks (Figure 1B). The animals were from Mozhugongka County in Tibet. Genomic DNA was extracted via the MiniBEST Genomic DNA Extraction kit (Takara, Kyoto, Japan). The Annoroad^®^Universal DNA Library Prep Kit v2.0 (Illumina^®^, San Diego, CA, USA) was used to construct a sequencing library. The DNBSEQ-T7 platform (Beijing Institute of Genomics, Beijing, China) was used for genome sequencing, with a coverage depth of ≥10× per individual. High-quality raw data (HQRs) were obtained by filtering with Fastp (v0.20.0) (https://github.com/OpenGene/fastp, accessed on 8 June 2023) and aligning the sequences to the LU_*Bosgru*_v3.0 (GCA_005887515.1) reference genome. The genetic variations were annotated via GATK (v4.2.4.1).

A genome-wide selection sweep analysis (GWSA) was performed via the following two parameters: the F_ST_ and the θπ ratio. A sliding window with a length of 40 Kb and a step size of 20 Kb was used. The candidate selective regions were defined as the top 5% intersecting windows between the F_ST_ and the θπ ratio. The iHS was estimated via the R package rehh 2.0 [31,32]. Variants with an iHS score greater than |2| were considered loci under selection. The domesticated yak reference genome (LU_*Bosgru*_v3.0) was obtained from the ENSEMBL database (http://useast.ensembl.org, accessed on 7 March 2024), and gene annotation was performed via the call_gene() function in the R package HandyCNV (v1.1.7) [31]. The final CDGs of the GWSA were determined on the basis of the interaction of these three parameters.

The detection of runs of homozygosity (ROH) was conducted via Plink 1.9 software [33,34] with the following parameters: (1) a window threshold of 0.05, (2) a sliding window of 50 SNPs, (3) a maximum of 5 missing genotype calls per window, (4) a maximum of 3 heterozygote calls per window, (5) a maximum gap of 100 kb between 2 consecutive SNPs, (6) a minimum ROH length of 300 kb, and (7) a minimum of 50 SNPs within a ROH. The inbreeding coefficients were calculated based on ROHs (F_ROH_) with the following formula: $F_{ROH}=\frac{\sum L_{ROH}}{L_{genome}}$, where $\sum L_{ROH}$ is the sum of the lengths of ROH fragments on autosomes, and $L_{genome}$ is the sum of the physical lengths of autosomal genomes (2670.082 Mb, in accordance with the LU_*Bosgru*_v3.0 Genome Assembly). The ROH results were analyzed and visualized via the R package HandyCNV (v1.1.7). The high-frequency ROH region, also known as the ROH islands, was defined as the window with a sample size of 20% or more. The CDGs selected from the ROH islands were defined as the overlapping genes of GWSA and those within the ROH islands [35,36]. Gene ontology (GO) and the Kyoto Encyclopedia of Genes and Genomes (KEGG) enrichment analyses were performed via the gene list enrichment module of the KOBAS (http://bioinfo.org/kobas/, accessed on 7 May 2024) online website, with significance determined at *p*-values less than 0.05.

## 3. Results and Discussion

We obtained a total of 1,517,494 SNPs and 130,119 windows from 50 animal autosomes. Within these, we identified 2263 CDGs, such as *CDK2*, *RAB5B*, and *PMEL*, which were annotated as candidate selection intervals on the basis of the top 5% windows of F_ST_ (>0.107884) (Figure 2A) and the θπ ratio (>3.108696) (Figure 2B). Additionally, our iHS detection results revealed that in the black coat population, 69,747 selected SNPs were annotated with 2801 CDGs (Figure 2C), whereas in the brown coat population, 69,327 selected SNPs were annotated with 2834 CDGs (Figure 2D). By intersecting the abovementioned parameters, we identified a total of 648 selected CDGs in the black yak population, including *LEF1*, *STAT3*, and *NTRK2* (Appendix A). Among these, 472 genes were enriched in 235 KEGG signaling pathways (Figure 2E) (Appendix A) and 2810 GO terms (Appendix A), which included significantly enriched pathways related to pigment deposition, such as melanogenesis, the cAMP signaling pathway, and protein tyrosine kinase binding. Similarly, we identified 691 selected CDGs (e.g., *PDGFC*, *SLC9A*1, and *DTNBP1*) in brown yaks (Appendix A), with 668 CDGs enriched in 249 KEGG signaling pathways (Figure 2F) (Appendix A) and 3377 GO terms (Appendix A). Notably, two pathways (melanogenesis and transmembrane receptor protein tyrosine kinase activity) were significantly enriched in the brown population. Furthermore, we found that 513 CDGs were shared between the two coat color populations, with 178 and 135 privately selected genes in the brown and black populations, respectively (Figure 2G).

In detail, our results confirmed the relationship between a significant number of CDGs and pigmentation. Notably, certain genes contributed significantly to the enrichment of the melanogenesis pathway. For example, the *PLCB1* gene encodes a protein that catalyzes phosphatidylinositol 4,5-diphosphate (PIP2), and its relevance to coat color determination has been demonstrated in goats and sheep [37]. Another gene, *ADCY8*, encodes adenosine cyclase 8, which catalyzes ATP to generate cyclic AMP. Elevated levels of cAMP have been shown to increase the expression of crucial genes involved in melanin production, including *TYR, TYRP1*, and *DCT* [38,39]. The *CREB3L2* gene, known for its homologs in the cAMP protein family, is believed to regulate melanin synthesis through its involvement in cAMP signaling [11,40]. Additionally, *WNT11*, a member of the Wnt family, has been shown to play a significant role in the development of melanocytes. Studies have revealed that *WNT11* promotes the proliferation and differentiation of melanocytes by upregulating the transcription of microphthalmia-associated transcription factor (MITF) and regulating the expression of upstream and downstream genes to facilitate melanin synthesis [41,42,43]. Moreover, GLI family member 3 (*GLI3*) has been implicated as a modifier in the regulation of early melanocyte proliferation and differentiation [44], whereas *LRMDA* has been associated with melanocyte differentiation, with deficient expression resulting in albinism [45].

In addition, the present study identified a significant number of distinct selected genes in the group of black-coated yaks. These genes include lymphoid enhancer factor 1 (*LEF1*), which is a transcription factor involved in the Wnt signaling pathway that stimulates the transcription of tyrosinase-related genes by regulating MITF and thus impacts melanin synthesis [46]. The activator of transcription 3 (*STAT3*) has been demonstrated to facilitate the growth of melanoma cells through phosphorylation [47]. *NTRK2* primarily encodes BDNF/NT-3 growth factor receptors, also known as tyrosine kinase receptor B. A previous study indicated that fusion mutations in the *NTRK2* gene can cause abnormal activity in melanoma cells [48]. Tyrosine phosphorylation-regulated kinase 4 (*DYRK4*) indirectly affects melanin synthesis by participating in protein tyrosine phosphorylation via its kinase activity [49]. Developmentally downregulated 9 (*NEDD9*) has been identified as a highly expressed gene in human metastatic melanoma, enhancing the invasive and metastatic capacities of melanoma cells [50]. Additionally, *HTR4*, which encodes 5-hydroxytryptamine receptor 4, and its agonist inhibits PI3K/Akt/mTOR signaling in vivo, leading to reduced S6 phosphorylation and subsequently promoting apoptosis in melanoma cells [51]. Furthermore, the mitogen-activated protein kinase (MAPK) family represents a crucial signaling pathway influencing melanin synthesis and the production of pivotal downstream melanin synthesis enzymes [52]. The present study revealed unique members of the three-brother family (*MAP2K6*, *MAP3K9*, and *MAPK10*) in the black population, suggesting that MAPK pathway signaling may serve as a fundamental basis for the black phenotype of yaks.

This study revealed that certain specific genes, such as *SLC9A1* (Na+/H+ exchanger 1), in brown yaks act as membrane transporter proteins that help maintain the pH balance. The disruption of pH can impact melanin synthesis and facilitate the migration and diffusion of individual melanoma cells [53]. Additionally, dystrobrevin-binding protein 1 (*DTNBP1*), a unique gene directly involved in melanosome biogenesis in brown yaks, plays a critical role in the transport of proteins to melanosomes. Deficiencies in *DTNBP1* function can lead to the destruction of melanosomes [24,54,55]. *FRK* has been shown to regulate the stability of phase and tens homolog deleted from chromosome 10 (PTEN) proteins, inhibiting the proliferation and invasion ability of melanoma cells [56]. Peroxisome proliferator-activated receiver alpha (PPARA) has been found to have ketogenic effects, and the ketone bodies it produces have antiproliferative and pro-apoptotic effects on melanoma cells [57,58]. Studies have also demonstrated that the activation of the aromatic hydrocarbon receptor (AHR) pathway enhances tyrosinase activity, the cellular melanin content, and the transcriptional levels of key genes involved in melanogenesis [59]. 

In this study, a total of 7116 ROHs were identified in the two yak populations (Appendix A), with the majority of ROH lengths falling within the range of 300–500 Kb. Additionally, the brown yak population not only presented a greater number of ROHs than the black population but also presented a fourfold greater proportion of longer ROH fragments (>700 Kb) than black yaks. ROHs are commonly used to assess the inbreeding coefficient of yak populations [60,61]. Notably, the average genomic inbreeding coefficient (F_ROH_) in the brown population (F_ROH_ = 0.029) was greater than that in the black yak population (F_ROH_ = 0.019). This could be attributed to recent inbreeding events and intense artificial selection, including limited effective population size and nonrandom mating, among other factors [62]. 

Furthermore, a total of 73 (Appendix A) and 75 (Appendix A) ROH islands were detected in the black (Figure 3A) and brown Sibu yak populations (Figure 3B), respectively. In each population, 326 and 243 genes were annotated. Additionally, by combining the CDGs from the GWSA, a total of 21 candidate selected genes (e.g., *SPAG16*, *SLC4A10*, and *TCF25*) were identified within the ROH islands of the black Sibu yaks (Appendix A) (Figure 3E). Among these, 17 genes were enriched in 13 KEGG signaling pathways (Figure 3C) (Appendix A) and 210 GO terms (Appendix A), including the tyrosine catabolic process, Fanconi anemia pathway, and other significantly enriched pathways. With respect to the brown population, a total of 15 candidate-selected genes (e.g., *SPOCK1*, *UVRAG*, and *SKAP2*) were identified from the ROH islands (Appendix A) (Figure 3E). Among these genes, 14 were enriched in 38 KEGG pathways (Figure 3D) (Appendix A), and 196 were enriched in GO terms (Appendix A). Notably, some of the significantly enriched pathways, such as the positive regulation of adenylate cyclase activity and the regulation of the canonical Wnt signaling pathway, were found to be related to pigmentation.

ROH fragments formed through homologous recombination are potentially associated with several economic traits influenced by strong artificial and natural selection pressures [63,64]. This study facilitated the identification of CDGs for yak coat color by screening selected regions within the ROH islands. In black yaks, certain CDGs found on these islands are widely recognized as being involved in melanin synthesis and pigment cell migration. One such gene is the aromatic hydrocarbon receptor repressor (*AHRR*), which encodes a protein that indirectly affects melanin synthesis through the aromatic hydrocarbon receptor pathway [65]. Additionally, *TCF25*, which is located near *MC1R*, has been associated with wool color and the development of melanism in Tibetan sheep and ducks [66,67]. Similarly, *RPL3* has been identified as a CDG associated with peritoneal pigment deposition in broiler chickens through a genome-wide association analysis [68], Uronyl 2-sulfotransferase (UST) is involved in the impaired adhesion and migration of melanoma cells [69]. Coincidentally, a substantial number of selected genes from the ROH islands were also discovered in the brown yak population. Among them, *TRPS1* is believed to affect melanin levels in vitro and play a role in regulating human skin color [70]. Gumonji and AT-rich interaction domain containing 2 (*JARID2*) have been shown to mediate melanoma growth by regulating the Wnt signaling pathway [71]. Importantly, the autophagic tumor suppressor (UVRAG) plays a crucial role in melanin formation during tanning reactions. UVRAG promotes melanin particle transport by interacting with lysosomal-associated organelle complex 1 (BLOC-1) and simultaneously participates in the tanning response dominated by α-melanocyte-stimulating hormone (α-MSH), causing skin pigmentation to resist ultraviolet radiation [72,73]. These findings suggest that brown yaks may gradually develop a defense and repair mechanism against UV damage to compensate for the risk of fur color defects, with black fur being particularly resistant to UV rays.

The activation of *MC1R* subsequently leads to tyrosinase activity, and the *TYR* gene family plays a crucial role in the efficiency of eumelanin production, which is essential for melanin production [11]. Research has confirmed that the *MC1R* recessive mutation in *Mus musculus* molossinus results in a yellow-brown color [74], whereas the *MC1R* recessive genotype in Huskies leads to a white color [75]. Additionally, recessive homozygous combinations of the *TYRP1* gene lead to a brown coat [14,15,16]. Although we did not identify these well-known key genes (such as *MC1R* and *TYRP1*) that contribute to the bright-colored coat phenotype (BCC), as expected, this study revealed that the brown Sibu population of yaks shows a greater degree of inbreeding than black animals do. This finding suggests the presence of novel recessive gene loci that determine the BCC in yaks among the CDGs we obtained. These results provide strong evidence for further speculation and validation of the genetic mechanism underlying the brown coat in yaks.

## 4. Conclusions

In summary, this study identified several pathways related to coat color through genome-wide selection signal analysis. These pathways include melanogenesis, melanocyte differentiation, melanoma, and melanosome organization, among others. The CDGs associated with these pathways serve as important theoretical references for understanding the genetic basis of yak coat color and environmental adaptation.

## Figures and Tables

**Figure 1 animals-14-02458-f001:**
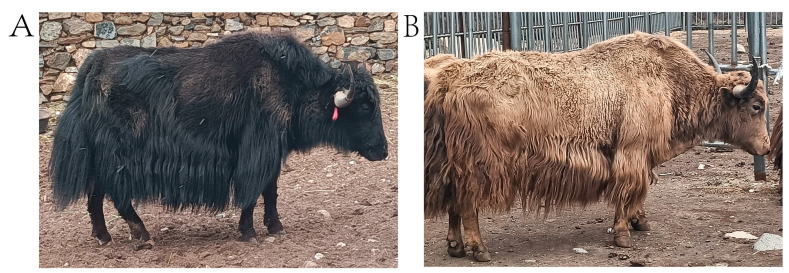
Different coat color phenotypes of Sibu yaks. (**A**) A side-view photograph of a black Sibu yak. (**B**) A side-view photograph of a brown Sibu yak.

**Figure 2 animals-14-02458-f002:**
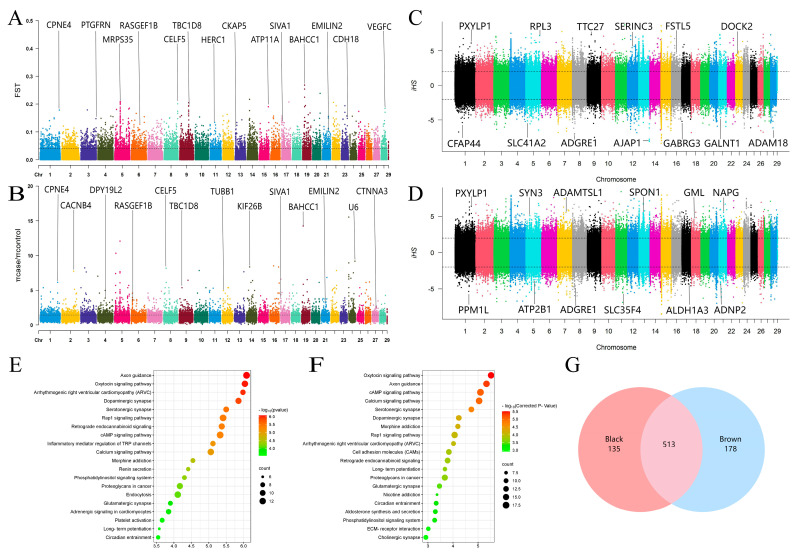
Results of yak coat color selection sweep analysis and KEGG pathway analyses of candidate selection genes. (**A**) Manhattan map of the F_ST_ analysis results comparing the black and brown yak groups. Each point represents an SNP, and the black dashed line indicates the top 5% threshold. (**B**) Manhattan map of the θπ ratio analysis results comparing the black and brown yak groups. Each point represents an SNP, and the black dashed line indicates the top 5% threshold. (**C**) Manhattan plot of the iHS selective signatures in the black yak group. The black dashed line illustrates the significant threshold > |2|. (**D**) Manhattan plot of the iHS selective signatures in the brown yak group. The black dashed line illustrates the significant threshold > |2|. (**E**) CDG functional enrichment analysis of the black yak group. Each circle represents a KEGG pathway. (**F**) CDG functional enrichment analysis of the brown yak group. Each circle represents a KEGG pathway. (**G**) Venn diagram of the interacting candidates in the black and brown yak groups with F_ST_, θπ ratio, and iHS parameters.

**Figure 3 animals-14-02458-f003:**
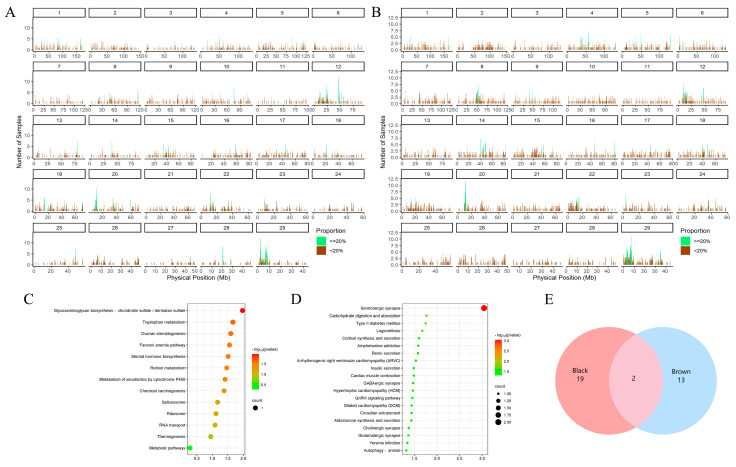
ROH results and enrichment analysis plots for different coat color groups. (**A**) ROH distribution map of the black yak group; green represents ROH islands. (**B**) ROH distribution map of the brown yak group; green represents ROH islands. (**C**) CDG functional enrichment analysis of the ROH islands of the black yak group. Each circle represents a KEGG pathway. (**D**) CDG functional enrichment analysis of the ROH islands of the brown yak group. Each circle represents a KEGG pathway. (**E**) Venn diagram of the interacting candidates in the ROH islands of the black and brown yak groups.

## Data Availability

The SNP genotype dataset for all individuals has been uploaded to the public database Genome Variation Map (https://bigd.big.ac.cn/gvm/getProjectDetail?Project=GVM000778, accessed on 17 June 2024).

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
