# Peer review of "Genome-Wide Selection Sweep Analysis to Identify Candidate Genes with Black and Brown Color in Tibetan Sibu Yaks"

_animals, 2024, doi:10.3390/ani14172458_

Round 1

Reviewer 1 Report

Comments and Suggestions for Authors

The manuscript is interesting and provides some new insights in the genetic mechanisms, which involved in coat color regulation in Tibetan Yaks. This species is important for local communities living in the hard-ti-get regions of the Tibet Mountains.

Comments

The photographs of the studied animals with different coat colors would be nice a supplement to the paper.

Figure 2A and B should be presented in higher resolution.

Please provide information whether different approaches have resulted in the identification of shared (common) genes, which might be proposed as functional candidates for coat color trait in Yaks.

The Introduction is very succinct. The authors may add some interesting details. For example, the roles of the Yaks for the supporting livelihood of local communities.

Please provide information whether local communities use the Yak`s wool in their livelihood.  Which coat variation is valued by local people?

Comments on the Quality of English Language

 Minor editing of English language required

Reviewer 2 Report

Comments and Suggestions for Authors

Beginning in line 40, more detail needs to be written on the issues of eumelanin (black or brown) versus pheomelanin (red or yellow). This is somewhat confused in the presentation, and is of fundamental importance. "Brown" animals can be either eumelanic or pheomelanic, and the two are often confused phenotypically but are quite different from genetic causation.

The criticism of other studies in line 56 may not be totally warranted.

More direct discussion should be included on single genes well-known to account for most color variation in livestock, such as MC1R and TYRP1.

The results on the relative levels of inbreeding are interesting, and point strongly to a recessive genetic mechanism for brown coat color. If this is true, then the segregation could either be at MC1R (and the brown is therefore likely pheomelanin) or TRYP1, in which case it is likely eumelanin. Those both exist in cattle, and this should be discussed further.

Reviewer 3 Report

Comments and Suggestions for Authors

The Authors performed a whole genome comparative analysis of two different yaks coat color phenotypes. They selected 50 unrelated Sibu yaks (25 brown and 25 black coat color) to identify candidate genes related to the coat color phenotype.

The study is interesting and I have only few remarks that the Authors can consider improving the manuscript.

Pay attention to the author guidelines: results and discussion have to be separated.

Line 64: please change “The venous blood of 50 (25 brown and 25 black coat color) adult Sibu yaks with nonblood relationship within three generations were collected.” with “The venous blood of 50 (25 brown and 25 black coat color) adult Sibu yaks unrelated individual within three generations were collected.”

Line 73: please change “The wide-genome selection sweep analysis (WGSA) was displayed by two parameters (FSTand π ratio) with 40 Kb length size…..” with “The wide-genome selection sweep analysis (WGSA) was displayed by two parameters, FSTand π ratio, with 40 Kb length size…..”.

Lines 75-76: please change “The iHS (integrated haplotype score) was estimated…” with “The integrated haplotype score (iHS) was estimated…..”.

Line 99: please change “….(CDGs)” with “…candidate genes (CDGs)”.

Line 137: please change “Furthermore, GLI3 (GLI Kruppel family member 3) is involved as….”with “Furthermore, GLI Kruppel family member 3 (GLI3) is involved as….”.

Line 142: please change “….LEF1 (lymphoid enhancer factor 1)…” with “…..lymphoid enhancer factor 1 (LEF1)….”.

Line 144: please change “STAT3 (activator of transcription 3)…” with “The activator of transcription 3 (STAT3)…”.

Line 148: please change “DYRK4 (tyrosine phosphorylation regulated kinase 4)….” with “tyrosine phosphorylation regulated kinase 4 (DYRK4)…”.

Line 150: please change “…NEDD9 (developmentally down-regulated 9)….” with “…developmentally down-regulated 9 (NEDD9)….”.

Lines 164-166: please change “DTNBP1 is a unique gene directly associated with melanosome biogenesis observed in brown yaks, and the deficiency of the DTNBP1 (Dystrobrevin-binding protein 1) function can hinder…” with “Dystrobrevin-binding protein 1 (DTNBP1) is a unique gene directly associated with melanosome biogenesis observed in brown yaks, and the deficiency of the DTNBP1 function can hinder…”.

Lines 168-170: please change “PTEN (phase and tense homolog deleted from chromosome 10) protein [43]. PPARA (peroxisome proliferator activated receiver alpha) has been…..” with “Phase and tense homolog deleted from chromosome 10 (PTEN) protein [43]. Peroxisome proliferator activated receiver alpha (PPARA) has been…..”.

Line 172: please change “AHR (aromatic hydrocarbon receptor) pathway can….” with “Aromatic hydrocarbon receptor (AHR) pathway can….”.

Lines 206: please change “the AHRR (aromatic hydrocarbon receptor repressor), which…” with ““the aromatic hydrocarbon receptor repressor (AHRR), which…”.

Line 213: please change “UST (uronyl 2-sulfotransferase)…..” with “uronyl 2-sulfotransferase (UST)…..”.

Line 216: please change “…JARID2 (jumonji and AT rich interaction domain containing 2) and PRC2 (polycomb responsive complex 2)” with “…jumonji and AT rich interaction domain containing 2 (JARID2) and polycomb responsive complex 2 (PRC2)”.

Line 218: please change “…UVRAG (autophagic tumor suppressor)….” with “…autophagic tumor suppressor (UVRAG)….”.

Round 2

Reviewer 2 Report

Comments and Suggestions for Authors

This paper is problematic. It focuses on many loci of marginal importance in color determination, and ignores others that are more central to that process. I suppose it could be re-titled to avoid a focus on coat color, and more on other differences between the black and brown populations.

Comments on the Quality of English Language

The English is understandable.
